# Early Maternal Caregiving Capacities in Highly Vulnerable, Multi-Problem Families

**DOI:** 10.3390/ijerph192316130

**Published:** 2022-12-02

**Authors:** Marije van der Hulst, Rianne Kok, Peter Prinzie, Eric A. P. Steegers, Loes C. M. Bertens

**Affiliations:** 1Department of Obstetrics and Gynecology, Erasmus MC, University Medical Center Rotterdam, Postbus 2040, 3000 CA Rotterdam, The Netherlands; 2Research Group Transforming Youth Care, The Hague University of Applied Sciences, 2521 EN The Hague, The Netherlands; 3Department of Psychology, Education and Child Studies, Erasmus University Rotterdam, 3062 DR Rotterdam, The Netherlands

**Keywords:** caregiving capacity, video-observation, vulnerable populations, early childhood, multi-problem families

## Abstract

Caregiving capacities may be an important link between multi-problem circumstances and adverse child development. This study aims to assess caregiving capacities and their correlations in highly vulnerable, multi-problem families in Rotterdam, the Netherlands. Caregiving capacity (overall, emotional and instrumental) was prospectively assessed in 83 highly vulnerable women using video-observations of daily caregiving tasks, six week postpartum. Supporting data were collected at three time points: at inclusion, six weeks after inclusion and six weeks postpartum, and these included psychological symptoms, self-sufficiency, problematic life domains, home environment, income, depression, anxiety and stress. Pregnancy- and delivery-related information was collected from obstetric care professionals. Maternal caregiving scores averaged below adequate quality. Mothers living in an unsafe home environment (*B* = 0.62) and mothers with more problematic life domains (≤3 domains, *B* = 0.32) showed significantly higher instrumental caregiving capacities. Other variables were not related to caregiving capacities. Caregiving capacity in this highly vulnerable population was below adequate quality. However, in most cases there was no significant association between caregiving and the variables related to vulnerability. This means that a potential association between vulnerability and caregiving capacities might be driven by the interaction between several problems, rather than the type or number of problems.

## 1. Introduction

Extensive research identified several risk factors for psychopathology and illness in offspring [1,2]. Children from low socio-economic backgrounds have a higher probability of perinatal mortality and morbidity [3]. Furthermore, they are more likely to report unhealthy behavior and suboptimal health in later life, suggesting an intergenerational transmission of adversity [3,4,5,6]. Although risk factors rarely present in isolation, the literature mainly focusses on the effects of single adversities on early child development. Families experiencing multiple risk factors also often experience problems in acquiring assistance from (public) services and often distrust professionals [7].

A review on child psychopathology in multi-problem families showed that accumulation of at least four risk factors results in a tenfold increase in probability of later dysfunction in children [7]. The effects were not only fueled by the accumulation of risk factors but also by persistence over time, type and configuration [7]. Previous research linked the severity of risk exposure through diminished parenting quality to impaired early child development [8]. Additionally, higher maternal distress is associated with harsher discipline towards their children [9]. Overall, caregiving capacities may be an important link between multi-problem circumstances and adverse child outcomes later in life. However, this association has not been extensively studied, especially early in the child’s life. 

Multi-problem families frequently cluster together in specific neighborhoods and in metropolitan areas. Within Rotterdam (the 2nd largest city of the Netherlands) 25% of children grow up in poverty, 19% live in families depending on welfare, 20% have parents with low educational attainment and 10% experience unemployment in the family [3,10]. To support maternal wellbeing and healthy child development a holistic approach was developed to integrate social and medical care for highly vulnerable families from pregnancy into early childhood, the “Mothers of Rotterdam” program [11]. Extensive home observations of mother–infant interactions were collected as early as 6 weeks postpartum. With this unique dataset we aim to assess the caregiving capacities and their correlates in highly vulnerable, multi-problem families. Our hypothesis is that caregiving capacities in this population are inadequate and related to indicators of vulnerability. 

## 2. Materials and Methods

Data originated from the Mothers of Rotterdam cohort study [11], a study on targeted social care for highly vulnerable pregnant women in Rotterdam, the Netherlands. In this program, highly vulnerable pregnant women received social care targeted to deal with their specific combination of faced adversities. Women were eligible for participation when they resided in Rotterdam, were pregnant during application for social care and were identified as highly vulnerable by a social care provider. Pregnant women facing a minimum of three adversities over at least two life domains were considered to be highly vulnerable. Application for social care included the details of the referring party and a list of adversities indicating the degree of vulnerability of the pregnant woman. Adversities on this list are known to influence the self-sufficiency, well-being and health of the woman or the health of her unborn child. Following application, a social care professional assessed care needs and study participation was discussed [11]. Women were excluded if not sufficiently skilled in Dutch, English, Arabic, Polish, Spanish or Turkish. Written informed consent was obtained from the mother herself and consent for children was obtained from all legal parents. The Mothers of Rotterdam study was approved in January 2016 (ref. no. MEC-2016-012) by the Erasmus Medical Center Ethics Committee. 

### 2.1. Data Collection

Data were collected from January 2016 to September 2019. Baseline measures were available for 321 women. The data of 210 women were excluded because of restrictions in parental consent. Another 28 were excluded because of insufficient data quality, resulting in a sample of 83 women.

Data were collected at three time points: at inclusion, six weeks after inclusion (only when still pregnant) and six weeks postpartum. Data consisted of questionnaires and a home observation six weeks postpartum. Social care providers also filled in questionnaires at inclusion. Delivery-related information was collected from obstetric care professionals. 

### 2.2. Outcome Variable

*Caregiving capacity* was assessed by the Dutch Infant Caregiving Assessment Scales (INCAS) [12,13]. The INCAS is a structured observational procedure to evaluate caregiving capacity in the early postpartum period in highly vulnerable populations. 

Mothers were observed at home 6 weeks postpartum with their infant during four caregiving tasks: bathing, dressing, feeding and changing a diaper. Observations were recorded from preparation until completion and rated on 13 scales ranging from 0–4, with scores of ≥3 signifying adequate caregiving capacities. Scores were aggregated into one overall score and in two dimensional scores: emotional caregiving and instrumental caregiving. Emotional caregiving encompasses maternal affection, quality of mother–infant interaction, maternal empathy and emotion regulation ability, maternal ability to adapt to infant’s needs and mothers’ mentalization skills. Instrumental caregiving promotes safety and describes consistency and the material sufficiency of the caregiving environment [12,13]. 

All videotapes were independently rated by two trained and supervised coders. Coders were blind to all other information on the mother–infant dyad. Single measure absolute agreement intraclass correlation was 0.726 on average (range 0.603–0.854) and showed a Cronbach’s *α* = 0.88 overall: 0.81 for emotional caregiving and 0.83 for instrumental caregiving. 

### 2.3. Determinants

*Psychological symptoms:* Rated by social care providers using selected items of the Health of Nation Outcome Scales [14]: aggression/over-activity, self-harm, substance abuse, hallucinations/delusions, depressed mood and other mental/behavior problems, from 0 (no problem) to 4 ((very) severe problem). Psychological symptoms were considered present when one or more of these items scored 2 or higher. 

*Self-sufficiency:* reported by social care providers using the self-sufficiency matrix. This matrix evaluates the participant’s self-reliance and functioning in 11 life domains, on a scale from 1 (in crisis) to 5 (thriving), with scores of 4 and higher representing adequate self-sufficiency [15]. The 11 domains were aggregated in an overall sum score. 

Risk factors were scored upon study entry on the domains of pregnancy, housing, finance, work and education, children, health, social functioning and legal matters. Three items were deleted due to inapplicability. For the analyses, this was categorized into: ≤3 problem domains (reference), 4 problem domains and ≥5 problem domains.

The *home-environment* was observed six weeks postpartum by a researcher using an adaptation of the Infant Toddler Home Observation for Measurement of the Environment [16,17]. Only items relating to hygienic and (physically) safe home environments were used. 

Additionally, *insufficient income* for housing, clothing, food and medical care was assessed via four questions (range “poor” to “very well”) answered by the mother. Answers were aggregated into “insufficient” and “sufficient” funds for all four domains.

*Emotional state* was measured in mothers with the Depression, Anxiety and Stress Scales Short (DASS 21) at all three time points of this study. The severity of symptoms regarding depression, anxiety and stress in the previous week is assessed in 21 questions, on a scale of 0–3, with higher scores reflecting more severe symptoms. Scores were summarized over the three domains [18,19]. In the analyses, a difference score was calculated between the two last measurements, one during pregnancy and one six weeks postpartum. 

### 2.4. Descriptive Variables

*Pregnancy- and delivery-related information* was extracted from the care files the obstetric care professionals, including parity, child sex, preterm or term birth and appropriate or low birth weight. 

*Cognitive ability* was assessed with the Abbreviated Nine-Item Raven’s Standard Progressive Matrices [20], filled out by the mothers. Cognitive functioning was categorized in two groups: >25th percentile (average to well above average) and ≤ 25th ((well) below average). 

Social care providers reported on the following: *maternal age* (in years), *personal relationship status* (single versus in a relationship with the father of the (unborn) child), *place of residence* (living in a deprived or non-deprived neighborhood according to the Netherlands Institute for Health Services Research (NIVEL)) [21] and *current smoking and substance abuse* (yes/no). 

### 2.5. Missing Data

Data were missing for income (3 participants (4%)), for cognitive ability (13 participants (16%)) and for emotional wellbeing (16 participants (19%)). Data on parity and birthweight were unavailable for 21 (26%) and 26 participants (32%), respectively. All other determinants had ≤1% missing data. Missing data were not imput. 

### 2.6. Statistical Analyses

The characteristics of the study cohort were summarized as means (M) with standard deviations (SD) or as absolute numbers and percentages. 

Linear regression analyses were performed to build two models with the three types of caregiving capacities as outcome (overall, emotional and instrumental caregiving capacity). The correlation between maternal age and the three outcomes was tested. Because of the non-significant and small level of correlation, maternal age was not included in the models. Model 1 focused on relatively stable circumstances, in which caregiving capacities were modeled using stepwise forward linear regression, consisting of four consecutive blocks: (1) psychological symptoms; (2) self-sufficiency; (3) risk factors; and (4) home-environment. Model 2 focused on more variable circumstances, modelling caregiving capacities as function of the difference scores on depression, anxiety and stress. Collinearity was tested in all models using the VIF estimate, and with all estimates being between 1.00 and 1.88, collinearity was considered not to be present. 

Two-sided *p*-values < 0.05 were considered to indicate statistical significance. All analyses were performed in IBM SPSS version 27 [22].

## 3. Results

The comparisons of the current and complete Mothers of Rotterdam study sample showed that mothers in the present study more often had a single relationship status, an unsafe home environment and resided in a deprived neighborhood (Table 1). On the other hand, mothers in the present sample had a lower frequency of psychological symptoms, more risk factors and insufficient income over all four categories. Regarding all other variables, the samples were comparable.

On average, mothers were 27 (SD = 5.5) years old, with around half living in a deprived neighborhood (47%) and living with their partner (51%). The majority (60%) of the mothers had below to well below average cognitive abilities. 

Roughly one third of the participants experienced psychological symptoms, mainly related to depression (19%) and stress (22%). Overall self-sufficiency averaged just below adequate, ranging from insufficient scores on income to self-sufficient scores within the legal domain. Most mothers had problems in five or more life domains (40%), with finance (88%), social functioning (81%) and health (76%) as the most frequently indicated problem areas. The home environment was considered unhygienic in 17% or unsafe in 15%. One third indicated insufficient income to provide for housing and clothing and 17% indicated as having insufficient income for food and medical care. 

From pregnancy to postpartum, depression decreased from 10.0 (SD = 8.6) to 8.2 (SD = 7.8), from mild to normal symptoms. Additionally, anxiety decreased from 8.4 (SD = 7.7) to 6.8 (SD = 6.5), from mild to normal symptoms. Stress decreased from 13.7 (SD = 9.5) to 10.9 (SD = 8.6), both in the normal range. All changes were not statistically significant.

On average, caregiving capacities were rated as below adequate caregiving quality, with overall caregiving capacities around 2.5 (SD = 0.6). Emotional caregiving capacities were lower (M = 2.2, SD = 0.7), whereas instrumental caregiving capacities were close to the cut-off for adequate care (M = 2.9, SD = 0.5). Considering emotional caregiving, mothers showed a less positive affect towards their infant. Interactions were more often misattuned to the infants’ needs, for example by intrusive touching or talking. Mothers showed limited empathy by not responding or dismissing the infant’s signals. On average, mothers handled infants more often like an object or were rough, without showing concern for the experience of their infant. In addition, mothers more often followed their own routine, without adapting to the cues of their infant, such as signs of discomfort or sleepiness. Mothers showed difficulty regulating infant’s negative emotions. Mentalization was generally low in this sample, with little evidence of acknowledgement, or incorrect perceptions, of the infant’s feelings, thoughts or interests. Instrumental caregiving capacities were below adequate caregiving quality as well. The observed quality was inconsistent, specifically regarding holding the infant, protection from harm and provision of basic needs. Inadequate holding was observed, e.g., when mothers let the infant slip in the bath, lifted their infant without adequate support of the neck and head and/or pulled limbs of their infant. An inadequate level of protection was reflected in neglectful or unsafe behavior, such as leaving the infant unsupervised or showing inadequate responses to signs of illness. Mothers were often unable to meet the basic physical and cognitive needs of the infant, such as using creme or lotion, a suitable bottle (if bottle-fed), clean fitting clothes and stimulation toys.

The associations were tested between caregiving capacities and either relatively stable circumstances, such as psychological symptoms, self-sufficiency, risk-factors and the home-environment in Model 1, or the more variable circumstances related to depression, anxiety and stress in Model 2. Most variables in Model 1 were not associated with caregiving capacities. Nevertheless, an unsafe home environment was significantly related to higher overall caregiving capacities (*B* = 0.52) and higher instrumental caregiving capacities (*B* = 0.62) (Table 2). Additionally, mothers with five or more problematic life domains showed significantly higher instrumental caregiving capacities, compared to ≤3 problematic domains (*B* = 0.32). Model 2 showed caregiving capacity to be unrelated to the difference scores on depression, anxiety and stress (Table 3).

## 4. Discussion

The results show that maternal caregiving capacities in our highly vulnerable families averaged below adequate caregiving quality, with the lowest scores in the domain of emotional caregiving capacities. Caregiving capacities were related to safety of the home environment and the number of problematic life domains. Mothers living in an unsafe home environment showed significantly higher overall and instrumental caregiving capacities, compared to mothers living in a safe home environment. Additionally, mothers with five or more problematic life domains showed significantly higher instrumental caregiving capacities to those with ≤3 problematic domains. Changes in depression, anxiety and stress were not related to their caregiving capacities. 

Emotional caregiving capacities were below adequate caregiving quality for almost all women. Prolonged exposure to stress could play a role. Psychological symptoms were present in 40% of the mothers, 60% have problems on four or more life domains and 30% experience insufficient income to pay for housing or clothes, all known to increase stress [23]. Previous research shows that long lasting stress negatively impacts the quality of caregiving, particularly more intrusive and hostile behavior and lower levels of sensitivity [23,24]. The levels of acute stress in our sample decreased from severe during pregnancy to moderate postpartum. However, overall caregiving capacity was not related to mother-reported acute stress levels nor chronic stress levels reported by the social care provider. Possibly, our study sample was overall too homogeneous in participants’ level of vulnerability to show these associations. Perhaps different or more extensive measures of stress are needed to study this association. A safe and nurturing environment is essential for children’s socio-emotional and cognitive development, and specifically, high-quality emotional caregiving supports resilience and is an important protective factor against the negative impact of adversities [8].

Instrumental caregiving seemed to be less affected. A possible explanation could be maternity care, which is received by approximately 95% of the mothers in the Netherlands [25,26]. Maternity care is mainly focused on information provision, practical caregiving skills, breastfeeding and the prevention and identification of health problems in mother and child [26]. Although emotional aspects of caregiving are also addressed, instrumental care is the main focus of maternity care. Another explanation may be that all mothers in our sample received some sort of support from the municipality, at least directly influencing the provision of physical basic needs but possibly other areas as well. Furthermore, the Netherlands has a wide array of support options, ranging from benefits to material support. Nevertheless, the majority of mothers indicated having insufficient income to provide for basic needs, such as housing and clothes.

Analyses on the association between relatively stable circumstances and caregiving showed that mothers with five or more problematic domains had better instrumental caregiving capacity compared to those with ≤3 problematic domains. Additionally, mothers living in an unsafe home environment showed significantly higher overall and instrumental caregiving capacities, compared to mothers living in a safe home environment. Although seemingly counterintuitive, these results indicate that it might be easier to adjust instrumental caregiving to adverse circumstances. Furthermore, more life problems and an unsafe home environment might make mothers more aware of their vulnerable situation and could motivate them to compensate for these circumstances in their caregiving. However, these hypotheses warrant further investigation.

Based on the previous literature, we expected an association between caregiving capacities and vulnerability [23]. However, we did not find this in our sample. One possible explanation could be that inadequate caregiving capacities are related to overall multi-problematic circumstances, rather than individual indicators of vulnerability. These multi-problematic circumstances leave mothers with little headspace to adequately care for their infant [23]. Suggesting that a potential association between vulnerability and caregiving capacities might be driven by the interaction between several problems, rather than the type or number of problems. Additionally, we did not include mothers with fewer than two problematic domains, thereby potentially missing contrast. Another explanation might be that we did not measure relevant variables associated with vulnerable circumstances. For example, we did not measure protective factors, such as resilience and social support, which could have influenced the individual associations between vulnerable circumstances and caregiving capacities [27]. Furthermore, psychological symptoms, the number of problematic life domains and self-sufficiency were not measured through self-reporting but reported by the social care provider. These instruments required some training and were therefore unsuitable for administration in mothers [14]. Introducing additional questionnaires to the mothers was not possible, as this study is part of the extensive Mothers of Rotterdam study and would increase the load of participation considerably. 

Moreover, although it is fairly unique to have such extensive data on caregiving capacities in this vulnerable group, the modest sample size could have restricted statistical power to detect significant relationships. Future studies are needed to confirm our findings and help identify potential mechanisms underlying the found associations.

An important limitation was the lack of consent in the majority of participants in the overall Mothers of Rotterdam sample. However, this is not entirely surprising. Vulnerable parents are known to be distrustful of (social) care providers and researchers [28,29]. Furthermore, some mothers did not live in their own house and did not have the permission of the occupant (often a family member or friend) for video-observations in the residence. Since the current study sample was overall less vulnerable compared to the overall study sample, caregiving capacities might even be less adequate in the overall study population.

A noteworthy observation was the absence of or poor language command during the video-observation. This may be the result of feeling awkward in the observational situation, language difficulties or cultural differences [30], and thus resulted in lower scores of emotional caregiving capacity, most specifically regarding mentalization. We attempted to maximize language use by observing mothers in a familiar environment and by emphasizing that caregiving should be as normal as possible, including speaking to their infant in their native language, in which case we used translations. 

Despite the limitations, this study reports on a unique and relatively large sample of multi-problem vulnerable mothers and combines data from multiple sources, timepoints and methods. Given the known difficulties in reaching vulnerable populations, this rich dataset on vulnerable mothers provides a unique insight into this otherwise underrepresented population. More in-depth explorations of how adversity affects caregiving capacity are needed to identify potential causal mechanisms and to inform and improve interventions.

## 5. Conclusions

This study shows the detrimental impact of vulnerability on caregiving capacities in early life. The fact that inadequate caregiving capacity was already observed in the first months after birth underlines the importance of early signaling and early (preventive) interventions in this group of highly vulnerable families. Caregiving interventions should be dedicated to improving emotional care. Especially since adequate caregiving supports resilience and has the potential to protect against the impact of adversities.

## Figures and Tables

**Table 1 ijerph-19-16130-t001:** Characteristics of the present study sample compared to the overall Mothers of Rotterdam dataset.

	Present Sample (N = 83)	Complete Dataset (N = 404)
INCAS		
Total (mean(SD))	2.53 (0.57)	Not applicable
Emotional (mean(SD))	2.17 (0.70)	Not applicable
Instrumental (mean(SD))	2.94 (0.54)	Not applicable
Model 1: Stable circumstances		
- Psychological symptoms Missing	31 (37.3) 1 (1.2)	172 (42.6) 43 (10.6)
Self-sufficiency (mean (SD))	42.5 (4.96)	42.14 (5.60)
Life domains:		
- ≤3 domains	25 (30.1)	109 (27.0)
- 4 domains	25 (30.1)	99 (24.5)
- ≥5 domains	33 (39.8)	196 (48.5)
Home environment		
Unhygienic home environment	14 (16.87)	27 (17.8) ^†^
Unsafe home environment	12 (14.5)	10 (6.6) ^†^
Insufficient income for:		
Housing	28 (33.7)	138 (55.2) ^‡^
Clothing	27 (32.5)	112 (44.8) ^‡^
Food	14 (16.9)	63 (25.2) ^‡^
Medical care	14 (16.9)	76 (30.4) ^‡^
Missing	3 (3.6)	
Model 2: Variable circumstances		
Depression (mean (SD))		
- During pregnancy	10.04 (8.56) ^§^	23.65 (9.39) ^¶^
- Postpartum	8.16 (7.82) ^§^	20.44 (7.99) ^¶^
- Difference score	2.71 (7.48) ^§^	−3.21 (7.87) ^¶^
Anxiety (mean (SD))		
- During pregnancy	8.36 (7.66) ^§^	22.99 (8.17) ^¶^
- Postpartum	6.77 (6.48) ^§^	19.93 (6.86) ^¶^
- Difference score	2.25 (6.55) ^§^	−3.06 (6.58) ^¶^
Stress (mean (SD))		
- During pregnancy	13.68 (9.48) ^§^	20.44 (7.99) ^¶^
- Postpartum	10.85 (8.64) ^§^	23.65 (8.58) ^¶^
- Difference score	3.85 (8.13) ^§^	−3.65 (8.47) ^¶^
Pregnancy- and delivery-related information		
- Male child	53 (63.9)	Not available
Primiparous	32 (38.6)	Not available
- Missing	21 (25.3)	Not available
Preterm birth	5 (6.0)	24 (8.3) ^#^
Small for gestational age	11 (13.3)	Not available
- Missing	26 (31.3)	Not available
Background variables		
Maternal age (mean (SD))	27.31 (5.45)	27.93 (6.09)
Cognitive abilities: (well) below average	50 (60.2)	193 (47.77)
-Missing	13 (15.7)	130 (32.18)
Personal relationship status: Single	40 (48.2)	159 (39.36)
- Missing	1 (1.2)	43 (10.64)
Living in a deprived neighborhood	39 (47.0)	161 (39.9)

Data are presented as number with frequencies unless stated otherwise. ^†^: N = 150, since most participants were not willing to participate in a home visit by the researcher; ^‡^: N = 250 due to missing data; ^§^: N = 72 during pregnancy, N = 79 postpartum due to missing data and for N = 68 it was possible to calculate a difference score; ^¶^: N = 141, since most participants were not willing to participate in a video-observation; ^#^: N = 289 due to missing data.

**Table 2 ijerph-19-16130-t002:** Maternal caregiving capacity and relatively stable circumstances.

	Total Caregiving			Emotional Caregiving			Instrumental Caregiving		
Model	B	Std. Error	t	B	Std. Error	t	B	Std. Error	t
Model 1									
Psychological symptoms	−0.18	0.13	−1.39	−0.23	0.16	−1.44	−0.13	0.13	−1.03
Constant	2.82	0.22	12.75 *	2.54	0.27	9.31 *	3.15	0.21	14.82 *
Model 2									
Psychological symptoms	−0.26	0.14	−1.86	−0.32	0.17	−1.85	−0.20	0.14	−1.47
Self-sufficiency	0.02	0.01	1.48	0.02	0.02	1.34	0.02	0.01	1.33
Constant	2.08	0.55	3.82 *	1.72	0.68	2.54 *	2.51	0.53	4.78 *
Model 3									
Psychological symptoms	−0.23	0.14	−1.57	−0.30	0.18	−1.65	−0.15	0.14	−1.08
Self-sufficiency	0.03	0.02	1.83	0.03	0.02	1.52	0.03	0.01	1.90
Life domains: 5 or more domains	0.23	0.17	1.37	0.17	0.21	0.80	**0.32**	**0.16**	**2.00 ***
Life domains: 4 domains	0.15	0.17	0.88	0.08	0.21	0.40	0.23	0.16	1.48
Constant	1.61	0.65	2.47 *	1.36	0.81	1.69	1.85	0.62	3.00 *
Model 4									
Psychological symptoms	−0.19	0.14	−1.35	−0.27	0.18	−1.48	−0.11	0.13	−0.82
Self-sufficiency	0.02	0.02	1.04	0.02	0.02	0.83	0.02	0.01	1.15
Life domains: 5 or more domains	0.24	0.17	1.41	0.17	0.22	0.79	**0.33**	**0.15**	**2.17 ***
Life domains: 4 domains	0.07	0.17	0.44	0.03	0.21	0.16	0.14	0.15	0.91
Insufficient income housing	0.09	0.14	0.66	0.01	0.17	0.06	0.19	0.12	1.53
Insufficient income clothing	0.26	0.14	1.82	0.32	0.18	1.74	0.19	0.13	1.49
Insufficient income food	−0.00	0.22	−0.00	0.14	0.28	0.48	−0.16	0.20	−0.81
Insufficient income medical care	−0.14	0.20	−0.71	−0.29	0.26	−1.11	0.02	0.18	0.09
Unsafe home environment	**0.52**	**0.20**	**2.64 ***	0.44	0.25	1.72	**0.62**	**0.18**	**3.40 ***
Unhygienic home environment	−0.05	0.17	−0.27	−0.03	0.21	−0.12	−0.07	0.15	−0.47
Constant	0.84	0.78	1.08	0.80	1.01	0.80	0.88	0.72	1.23

N = 83 for all models. * significant < 0.05; Explained variance per outcome; Model 1 R^2^ = 0.024/R^2^ = 0.026/R^2^ = 0.014; Model 2 R^2^ = 0.028/R^2^ = 0.028/R^2^ = 0.018; Model 3 R^2^ = 0.041/R^2^ = 0.031/R^2^ = 0.058; Model 4 R^2^ = 0.199/R^2^ = 0.138/**R^2^ = 0.272.**

**Table 3 ijerph-19-16130-t003:** Maternal caregiving capacity and relatively variable circumstances.

	Total Caregiving			Emotional Caregiving			Instrumental Caregiving		
	B	Std. Error	t	B	Std. Error	t	B	Std. Error	t
Depression	−0.00	0.01	−0.44	−0.00	0.01	−0.21	−0.01	0.01	−0.63
Anxiety	0.02	0.01	1.50	0.03	0.02	1.59	0.01	0.01	0.93
Stress	0.00	0.01	0.20	−0.00	0.02	0.13	0.01	0.01	0.62
Constant	2.53	0.07	37.52 *	2.14	0.09	24.26 *	2.99	0.06	48.74 *

N = 83 for all models. * significant < 0.05.

## Data Availability

The data presented in this study are available on request from the corresponding author. The data are not publicly available due to privacy issues of the participants in the study.

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
