# Peer review of "Early Maternal Caregiving Capacities in Highly Vulnerable, Multi-Problem Families"

_ijerph, 2022, doi:10.3390/ijerph192316130_

Round 1

Reviewer 1 Report

I would like to thank the Editor and the Authors for the opportunity to review the manuscript entitled: Early maternal caregiving capacities in highly vulnerable, multi-problem families

I find the contribution of relevant contents for the Journal and I appreciate very much the focus, so important to favor the child’s and her/his family wellness. However, I think that the manuscript, being a perspective contribution, needs some revisions.

Are the Authors sure about the statement “Mothers living in an unsafe home environment (B=0.62) and mothers with more problematic life domains (≤3 domains, B=0.62) showed significantly higher instrumental caregiving capacities” in the abstract?

Could the authors specify the minimum sample size necessary for the intended analyses?

Please, add the study’s internal consistency coefficient for each adopted instrument?

Could the Authors report test for collinearity?

Finally, I believe that identifying developmental processes that are disrupted by adverse early environments is crucial to plan more efficacious interventions. Yet, within a cumulative-risk approach it is unlikely to reveal these processes. In the discussion/limitations, could the authors refer to the importance of distinguishing between distinct types of environmental experiences? Indeed, different mechanisms may be involved with respect to different factors and this inevitably impacts intervention.

Reviewer 2 Report

The study “Early maternal caregiving capacities in highly vulnerable, multi-problem families” provide valuable insight from the conditions and caregiving abilities of vulnerable mothers. In particular, adapting a mixed-method approach is helpful with capturing unbiased observations. There are a number of points, however, that need to be addressed before the manuscript can be published:

Background:

1. Although a citation is provided, more context about “Mothers of Rotterdam” program can help with depicting the conditions; how these mothers/families were identified/what made them eligible? what kind of help is provided through this program? Also, were these mothers receiving help from any other organizations/charities? etc.

Sampling and recruitment:

2. As part of the method, it adds clarity to know how mothers were particularly recruited for the qualitative study (video recording)? were all of those covered under “Mothers of Rotterdam” approached or was it a snowballing process or other? Is the present sample representative of the mothers covered under this program?

Data analysis and determinants:

3. It seems that the videos were analyzed deductively? Were there any other observations that could lead to some inductive findings?

4. While number of factors, e.g. substance abuse, is aggregated under “Psychological symptoms”, is it possible to run the analysis for these variables individually to identify any heterogeneity?

5. Is it right to assume that you have controlled for mother’s age and other background variables in the analyses? please clarify in the manuscript.

Results presentation:

6. Table 1: the breaking rows (i.e. Model 1: Stable circumstances / Block 1: Psychological symptoms etc.) make the table unnecessary crowded and hard to read. I would suggest to remove these and present the characteristics in a cleaner and shorter format.

7. Is Age shown in Table 1 as mean value? What is the value in parentheses for this variable showing? Also, tables' font and style are inconsistent.

8. Tables 2 and 3: I would move the rows for constant to below the key variable for each stepwise model (as it is, Table 2 is particularly hard to read). For each model it is also important to report the number of observations (N).

Findings:

9. The counterintuitive relationship between Instrumental caregiving and Unsafe home environment as well as higher than 5 issues with life domain could be due to the material help that this particular families have received from various charities? Can you clarify that whether or not these families were further targeted by the social services?

10. Could there be the case that “Life domains: 5 or more domains” to be strongly correlated with “Unsafe home environment”? Also, these two variables were simultaneously introduced to the model 4? what might have happened if you added these one at the time?

11. It will be interesting to know the impact of mother’s age and education in the complete models (model 4)?

12. Another factor missing is that whether the family is refugee/recent immigrants who might be suffering from other traumas?

13. The information on pregnancy conditions i.e. whether or not the pregnancy was planned (that could indicate mother’s caregiving preparedness) are missing.

14. It is understandable that often, a qualitative study misses on using a large sample; however, this needs to be highlighted as part of the study limitations and the present inference might miss on statistical power due to the small sample size, hence, significant relationships can’t be detected?

Round 2

Reviewer 2 Report

My thanks go to the authors for addressing the concerns raised in the previous report thoroughly.